# Expression of the *ripAA* Gene in the Soilborne *Pseudomonas mosselii* Can Promote the Control Efficacy against Tobacco Bacterial Wilt

**DOI:** 10.3390/biology11081170

**Published:** 2022-08-04

**Authors:** Tao Zhuo, Shiting Chen, Dandan Wang, Xiaojing Fan, Xiaofeng Zhang, Huasong Zou

**Affiliations:** State Key Laboratory of Ecological Pest Control for Fujian and Taiwan Crops, College of Plant Protection, Fujian Agriculture and Forestry University, Fuzhou 350002, China

**Keywords:** *Pseudomonas mosselii*, *ripAA*, bacterial wilt, defense signaling pathway, control efficacy

## Abstract

**Simple Summary:**

Tobacco bacterial wilt caused by *Ralstonia solanacearum* brings large economic losses every year. Currently, an increasing number of biocontrol agents have been widely used in the control of this disease, but they cannot replace chemical agents, mostly due to the poorer control effect. Therefore, in this study, the avirulence gene *ripAA* from *Ralstonia solanacearum*, which determines incompatible interactions with tobacco plants, was introduced into the biocontrol agent *Pseudomonas mosselii* to increase the efficacy against *Ralstonia solanacearum*. The newly engineered strain can improve the systemic resistance and elicit a primary immune response of plants. Our research not only provides a new strategy for the genetic modification of biocontrol agents, in which a number of avirulence genes from the pathogen or plant can be tested to be expressed in different biocontrol agents to antagonize this plant disease, but also helps the study of the interaction between the phytopathogenic avirulence gene and the host.

**Abstract:**

The environmental bacterium *Pseudomonas mosselii* produces antagonistic secondary metabolites with inhibitory effects on multiple plant pathogens, including *Ralstonia solanacearum,* the causal agent of bacterial wilt. In this study, an engineered *P. mosselii* strain was generated to express *R. solanacearum ripAA*, which determines the incompatible interactions with tobacco plants. The *ripAA* gene, together with its native promoter, was integrated into the *P. mosselii* chromosome. The resulting strain showed no difference in antimicrobial activity against *R. solanacearum*. Promoter-LacZ fusion and RT-PCR experiments demonstrated that the *ripAA* gene was transcribed in culture media. Compared with that of the wild type, the engineered strain reduced the disease index by 9.1% for bacterial wilt on tobacco plants. A transcriptome analysis was performed to identify differentially expressed genes in tobacco plants, and the results revealed that ethylene- and jasmonate-dependent defense signaling pathways were induced. These data demonstrates that the engineered *P. mosselii* expressing *ripAA* can improve biological control against tobacco bacterial wilt by the activation of host defense responses.

## 1. Introduction

All major classes of plant pathogens interact with host plants in the “gene-for-gene” model in which a plant resistance (R) protein acts as an elicitor–receptor to directly or indirectly recognize the pathogen-derived *avr* gene product [1,2]. The defense reaction commonly occurs at infection sites to restrict pathogen growth and, in some instances, triggers a hypersensitive response (HR) [3,4]. Upon the recognition of an *avr* gene-harboring pathogen, host cells at the infection site generate an oxidative burst, producing reactive oxygen species, driving the cross-linking of cell-wall compounds, and leading to the expression of plant genes involved in cellular protection and defense [5,6]. In concert with defense responses, plant *R* gene-mediated pathogen resistance is extensively used for plant resistance breeding. For example, the transduction of the rice *Xa21* resistance gene into banana confers resistance to *Xanthomonas campestris* pv. *musacearum*, which causes the important banana *Xanthomonas* wilt disease in eastern and central Africa [7].

Devastating bacterial wilt caused by the *Ralstonia solanacearum* species complex is a major threat to more than 250 plant species in the tropics, subtropics, and other warm temperate areas [8]. As all tobacco plant cultivars are suitable hosts for *R. solanacearum* [9], bacterial wilt is one of the most important diseases in tobacco production areas. From 1989 to 1991, in a comprehensive investigation of 16 tobacco production provinces in China, bacterial wilt was found in all provinces except Heilongjiang and Jilin, characterized by high latitudes [10]. A high prevalence was found in southern China, especially Fujian, Guangdong, Yunnan, and Guangxi [10]. Over time, the disease has become more severe owing to the lack of effective prevention methods.

Like other plant pathogenic bacteria, *R. solanacearum* injects a number of effector proteins into host cells via a type III secretion system [11,12]. Dozens of genomes have indicated a meta-repertoire of effectors in *R. solanacearum* strains [13,14,15]. Among them, the Avr protein RipAA (AvrA) acts as a major host specificity factor recognized by *N*. *tabacum* and *N*. *benthamiana* [16,17]. The *ripAA* locus was first cloned from the *R. solanacearum* strain AW1 in a 2 kb DNA fragment determining tobacco incompatibility at the species level [18]. The transient expression of RipAA in tobacco leaves induces HR [16]. After infiltration into leaves, *R. solanacearum* strains containing the wild-type *ripAA* allele elicit an HR reaction from 16 to 24 h after inoculation, while strains containing a mutated allele cause chlorosis symptoms 36 to 72 h after inoculation, followed by necrosis 48 to 96 h after infiltration [16]. The wild-type *ripAA* allele is essential for *R. solanacearum* to induce HR or avirulence in tobacco [17]. *R. solanacearum* interacts with tobacco plant species in a “gene-for-gene” relationship depending on wild-type *ripAA*.

As *R. solanacearum* can survive for many years in the environment and invades hosts through plant roots [8], bacterial wilt disease has dramatic effects on contaminated fields owing to limited management measures. In the past few decades, great efforts have been made to reduce disease incidence by the application of antimicrobial agents [19]. We previously detected an environmental biological agent with promising control efficacy on tomato bacterial wilt [20]. Whole-genome sequencing analysis revealed that this agent was a *Pseudomonas mosselii* A1 strain harboring a type III secretion system, similar to the plant growth-promoting *P*. *fluorescens* [21]. To assess whether the *R. solanacearum* effector RipAA induces resistance to tobacco bacterial wilt, the *ripAA* gene was genetically expressed in *P*. *mosselii* A1 in this study. Our aim was to construct an engineered *P*. *mosselii* A1 strain harboring a *ripAA* gene with improved control efficacy in order to facilitate the development of a sustainable protection strategy against a soilborne disease, namely the tobacco bacterial wilt.

## 2. Materials and Methods

### 2.1. Bacterial and Plant Materials

The bacterial strains and plasmids used in this study are listed in Table 1. The *P*. *mosselii* strain was cultivated in Kings B medium containing 1.5% proteose peptone, 1.0% glycerol, 0.15% MgSO_4_, and 0.15% K_2_HPO_4_ at 28 °C [22]. *R. solanacearum* was isolated from a diseased tobacco plant in Sanmin, Fujian Province, China. a nutrient-rich (NB) medium containing 0.5% polypeptone, 0.1% sucrose, 0.3% yeast extract, and 0.6% beef extract was used to cultivate *R. solanacearum* at 28 °C [23]. *N*. *tobacum* NC89 was grown in a greenhouse at 28 °C with a photoperiod of 16 h of light (7.5 μmol/m^2^·s) and 8 h of darkness.

### 2.2. Integration of ripAA into the P. mosselii Chromosome

Three DNA fragments were cloned in pK19mobSacB to create a construct for integration. The primers used for molecular cloning are listed in Appendix A. The first 604 bp of the DNA fragment containing the 5′ end of a hypothetical gene was amplified from a *P. mosselii* A1 genomic DNA and cloned into pK19mobSacB at *Bam*HI/*Sal*I. An additional *Eco*RI cutting site was introduced in the reverse primer in front of *Sal*I. Second, the 677 bp 3′-end region of the hypothetical gene was cloned into *Eco*RI/*Sal*I. In this PCR amplification, a *Hin*dIII cutting site was introduced in the forward primer behind *Eco*RI. Finally, the DNA fragment containing *ripAA* and its promoter was amplified from an *R. solanacearum* GMI1000 genomic DNA and cloned at *Eco*RI/*Hin*dIII. The resulting construct contained the insertion sequence of the hypothetical gene interrupted by the *ripAA* gene. After the introduction of the construct into wild-type *P*. *mosselii* A1 via electroporation, a standard two-step homologous recombination procedure was used to isolate the marker-free *ripAA* insertion mutant [23].

### 2.3. Bacterial Confrontation Assay

The antagonistic activity of *P*. *mosselii* A1 on tobacco against *R. solanacearum* was conducted as described previously [20]. *P*. *mosselii* A1 and tobacco *R. solanacearum* were cultured in a Kings B and NB broth medium to OD600 = 1.5, respectively. To produce bacterium-containing plates, 1 mL of tobacco *R. solanacearum* cells was added to 100 mL of the NB agar medium. After the medium was solidified, 2 μL *P*. *mosselii* cells were spotted on each Petri dish and incubated for 3 days at 28 °C. The antimicrobial activity of *P*. *mosselii* was counted based on the inhibitory zone around bacterial colonies, and the diameter of the inhibitory zone was recorded using a ruler.

### 2.4. β-Galactosidase Activity Driven by the ripAA Promoter in P. mosselii A1

The first 276 bp promoter region of *ripAA* was amplified from the *R. solanacearum* GMI1000 genomic DNA (Appendix A) and cloned into a pLacZ-Basic vector (Clontech, Japan) at *Kpn*I/*Xho*I sites. Then, a DNA fragment including the promoter region and *lacZ* gene fusion was digested and moved into pBBR1MCS-5 via digestion with *Kpn*I and *Sal*I to generate pBB:PLacZ. Meanwhile, a 4630 bp DNA fragment containing only *lacZ* was generated from an empty pLacZ-Basic vector and cloned into pBBR1MCS-5 using the same cutting enzymes *Kpn*I and *Sal*I to generate pBB:LacZ. Finally, the generated pBB:PLacZ and pBB:LacZ constructs were separately introduced into *P*. *mosselii* A1 via electroporation for β-galactosidase activity analyses according to a standard procedure for the Miller assay [26]. For each strain, β-galactosidase activity was analyzed in Kings B or M63 culture conditions.

### 2.5. Semi-Quantitative RT-PCR

Total RNAs were isolated from *P*. *mosselii* A1 and AA1 cells grown in Kings B or M63 broth using a Trizol reagent (Invitrogen, Carlsbad, CA, USA) according to the manufacturer’s protocol. After contaminated gDNA was removed by treatment with RNase-free DNaseI (TAKARA, Otsu, Japan), 2 μg of purified RNA for each sample was used to synthesize cDNA using M-MLV Reverse Transcriptase (Promega, Madison, WI, USA) at 42 °C for 1 h. PCR parameters were as follows: 95 °C for 5 min followed by 30 cycles of 95 °C for 1 min, 60 °C for 30 s, and 72 °C for 30 s, and a final extension for 10 min at 72 °C. The expression of 16S rRNA was determined as a control to evaluate the relative amount of *ripAA* mRNA (Appendix A). For each PCR amplification, 8 μL of the product was loaded on a 1.5% agarose gel and visualized via ethidium bromide staining.

### 2.6. Controlling Tobacco Bacterial Wilt Using P. mosselii AA1

The cultured cells of *R. solanacearum*, *P*. *mosselii* A1, and *P*. *mosselii* AA1 were grown to the mid-logarithmic growth stage at 28 °C, harvested at 5000× *g* for 10 min, and resuspended to a final concentration of 1 × 10^8^ CFU/mL. The roots of 4-week-old *N*. *tobacum* NC89 plants were first wounded and then all inoculated with an *R. solanacearum* cell suspension at 5 mL per pot to make sure that the concentration of 3 × 10^6^ CFU/g in the soil of one pot. Afterward, tobacco plants were drenched with either *P*. *mosselii* A1 or AA1. For a negative control, 5 mL of water was applied to the tobacco roots instead of the antagonistic bacterium. All treated plants were grown at 28 °C with soil moisture of >90%. The disease development was recorded every day until all the control plants completely collapsed, according to the following the five-point rating scale (0–4), where 0 = no wilt symptoms, 1 = 1% to 25% leaves wilted, 2 = 26% to 50% leaves wilted, 3 = 51% to 75% leaves wilted; and 4 = 76% to 100% leaves wilted [20]. The disease index indicated the percent severity index (PSI), and the PSI was calculated according to the following formula: PSI = ∑(each rating scale × number of plants rated in the corresponding scale) ×100/(number of all plants rated × maximum scale of the scores) [27]. Each treatment included 8 plants and was performed in triplicate for statistical analysis.

### 2.7. High-Throughput RNA Sequencing

RNA sequencing was performed with GeneBang (Chongqing, China). Root samples were collected from 4-week-old *N*. *tobacum* NC89 plants 2 days after inoculation with 5 mL of a 1 ×10^8^ CFU/mL cell suspension of *P*. *mosselii* A1 or AA1. Total RNAs were extracted from roots using an RNeasy^®®^ Plant Mini Kit (QIAGEN, Shanghai, China) according to the manufacturer’s manual, and mRNAs were purified using poly-T oligo-attached magnetic beads. The concentration, purity, and integrity of RNAs were assessed using a NanoDrop^TM^ spectrophotometer (Thermo Scientific, Waltham, MA, USA) and an RNA 6000 Nano Kit with the Bioanalyzer 2100 system (Agilent Technologies, Santa Clara, CA, USA). A TruSeq RNA Library Preparation Kit (Illumina, San Diego, CA, USA) was used to generate RNA libraries, and the quality was assessed using the Agilent Bioanalyzer 2100 system. The plants treated with water were used as a negative control. For each treatment, triplicate experiments were performed for statistical analyses. Thereafter, nine libraries were constructed and were deep-sequenced on the Illumina HiSeq 4000 platform. The final reads were aligned to the complete reference genomic of Edwards 2017 in the Solanaceae plant genomics database network (ftp://ftp.solgenomics.net/genomes/Nicotiana_tabacum/edwards_et_al_2017 (accessed on 31 July 2017)). Genes with log2 fold change ≥ 1 and P ≤ 1 in the comparison between control and treatment groups were determined using the DESeq2 (version 1.12.3) package in R (version 3.3.2) and tested for significant differences using Cuffdiff v.2.2.156 and considered differentially expressed. Next, Fisher’s exact test and the false discovery rate (FDR) were used to evaluate each differentially expressed gene (DEG). KEGG analysis was performed using the differentially expressed genes.

### 2.8. Quantitative RT-PCR

Two micrograms of DeSeq2 RNA was used for reverse transcription (Promega, USA). All the primers used for qRT-PCR are listed in Appendix A. Assays were performed using the Bio-Rad CFX Connect^TM^ real-time system with SYBR premix Taq (BIO-RAD, Hercules, CA, USA). The PCR thermal cycle conditions were as follows: denaturation at 95 °C for 30 s and 40 cycles of 95 °C for 5 s, 55 °C for 30 s, and 72 °C for 10 s. The expression of EF1α was used as the internal control.

### 2.9. Accession Numbers

All the data of transcriptome were deposited with NCBI Sequence Read Archive (SRA) (https://www.ncbi.nlm.nih.gov/sra (accessed on 19 November 2018)). The accession numbers of root samples treated by H_2_O, *P. mosselii* A1, or *P. mosselii* AA1 from three biological repeats were SRR8205398, SRR8205405, SRR8205406, SRR8205399, SRR8205400, SRR8205402, SRR8205401, SRR8205403, and SRR8205404, respectively.

## 3. Results

### 3.1. Construction of an Engineered P. mosselii A1 Carrying the ripAA Gene

Owing to the necessity for incompatible interactions with tobacco plants, the *R. solanacearum* effector gene *ripAA* was genetically expressed in *P*. *mosselii* A1. A hypothetical gene located from 119,609 to 120,907 in the *P*. *mosselii* A1 chromosome (GenBank No. NZ_CP024159.1) was chosen as the insertion site to avoid any effects on bacterial growth or antimicrobial activity (Figure 1a). Three DNA fragments were inserted into the molecular cloning sites of the suicide vector pK19mobSacB, including the left and right parts of the hypothetical gene and a fragment containing the *ripAA* gene with the native promoter. The third fragment was inserted in the middle of the left and right parts of the hypothetical gene (Figure 1b). After the introduction of the resulting recombinant construct into wild-type *P*. *mosselii* A1, the *ripAA* gene with the native promoter was integrated into the chromosome following two steps of homologous recombination. The engineered *P*. *mosselii* strain was named AA1 and verified with PCR amplification (Figure 1c).

### 3.2. The ripAA Gene Is Transcribed in P. mosselii AA1

To determine whether the *ripAA* gene was transcribed in *P*. *mosselii* AA1, the promoter activity of *ripAA* was first examined in the *P*. *mosselii* A1 strain via fusion with the *lacZ* gene in a pBBR1MCS-5 vector. In contrast to the negative control harboring the *lacZ* gene without the promoter, the *ripAA* promoter activated *lacZ* gene expression in both Kings B and minimal medium M63. Under either culture medium, yellow colors were clearly observed from the strain carrying the *ripAA* promoter *lacZ* fusion 3 min after the addition of assay buffer containing 2-nitrophenyl-β-D-galactopyranoside (ONPG), and no color was detected from the control strain (Figure 2a). Quantitative analysis revealed that β-galactosidase activity driven by the *ripAA* promoter was dramatically higher than that of the control (Figure 2b). This suggested that the *ripAA* promoter was sufficient to drive the transcription of *ripAA* in *P*. *mosselii* A1. Semi-quantitative RT-PCR was subsequently performed to detect the transcriptional level of *ripAA* in the engineered strain. As shown in Figure 2c, the transcription of *ripAA* was detected in *P*. *mosselii* AA1 but not in *P*. *mosselii* A1 cultured in either Kings B or M63 broth. By contrast, the internal control 16S RNA was expressed with no difference. This demonstrated that *ripAA* was transcribed in *P*. *mosselii* AA1.

### 3.3. P. mosselii AA1 Retains Antimicrobial Activity on R. solanacearum

A bacterial confrontation assay was conducted on plates to evaluate whether the insertion of *ripAA* had an effect on antimicrobial activity. Like the wild-type *P*. *mosselii* A1, the engineered *P*. *mosselii* AA1 expressing *ripAA* showed antimicrobial inhibition activity on the growth of *R. solanacearum*. Inhibition zones were clearly observed around the colonies of both *P*. *mosselii* AA1 and A1 at 3 days post-inoculation (Figure 3a). The diameters of the inhibition zones were approximately 1.5 cm for both strains (Figure 3b). Statistical analysis indicated no difference between *P*. *mosselii* A1 and AA1, meaning that the expression of *ripAA* did not affect the antimicrobial activity of *P*. *mosselii* AA1.

### 3.4. P. mosselii AA1 Increases Control Efficacy on Tobacco Bacterial Wilt

*P*. *mosselii* AA1 was applied to root-wounded tobacco plants to assess control efficacy. In the negative control (water treatment), *R. solanacearum* resulted in wilt disease symptoms in tobacco plants at 3 days post-inoculation, whereas the plants receiving A1 and AA1 treatments showed wilt symptoms at 4 and 5 days post-inoculation, respectively (Figure 4). The plant disease index for the water-treated control increased to 92.9 at 15 days post-inoculation (Figure 4). The disease index for A1 treatment finally developed to 38.5 and that of the AA1 treatment developed to 29.4. In contrast to the A1 treatment, the plants receiving AA1 treatment exhibited a delay of 1 day for wilt symptom appearance; the final disease index was reduced by 9.1% (Figure 4).

### 3.5. Transcriptome Analysis of Tobacco Plants Treated with P. mosselii AA1

DeSeq2 was used to identify tobacco genes that were differentially expressed in tobacco plants treated with the AA1 strain. The transcription data for plants receiving AA1 treatment were analyzed by comparisons with both H_2_O and A1 treatments. In the comparison with the H_2_O treatment, 2497 genes were differentially expressed, including 1296 upregulated and 1201 downregulated genes (Figure 5a). In the comparison with the A1 treatment, 1277 genes were upregulated, and 1112 genes were downregulated. The genes that showed the same expression changes for the H_2_O and A1 treatments were recognized as potential genes affected by *ripAA* (Figure 5a). In total, 890 genes were upregulated, and 635 were downregulated in plants receiving AA1 treatment in comparison with both A1 and H_2_O (Figure 5a). The transcription levels of 10 genes from these differentially expressed genes were detected via qPCR, and the expression trend was consistent with the result of DeSeq2 (Appendix A). These 1525 differentially expressed genes were proposed to be specifically affected by *ripAA*.

Among the above-mentioned 1525 differentially expressed genes, 256 upregulated and 54 downregulated genes were classified into 31 functional pathways by the KEGG pathway enrichment analysis (Figure 5b). Among the 310 enriched genes, 79 genes were classified into several functional groups, implying the versatile biological functions of the genes. Furthermore, 89 genes were enriched in the metabolic pathway, which had the largest number of genes among 31 enriched pathways (Figure 5b). Four enriched pathways were upregulated by *ripAA*, including starch and sucrose metabolism, phenylpropanoid biosynthesis, pyrimidine metabolism, and the MAPK signaling pathway in plants (Figure 5b). Alanine, aspartate and glutamate metabolism, peroxisome, carotenoid biosynthesis, and glycosphingolipid biosynthesis pathways were downregulated. 

The ethylene and jasmonate signaling pathways were activated by AA1 expressing *ripAA*. The upregulation of 1-aminocyclopropane-1-carboxylate synthase (ACS) (Nitab4.5_0002381g0080.1) and the downregulation of methyl jasmonate esterase (MJAE) (Nitab4.5_0000382g0070.1) suggested that a metabolic increase in ethylene biosynthesis occurred in tobacco plants receiving AA1 treatment. Accordingly, downstream signaling transduction was activated, coupled with the elevated expressions of *efr1* and *etr1* (Figure 5c). For the JA signaling pathway, seven jasmonate ZIM domain-containing genes were downregulated.

## 4. Discussion

Numerous integrated measures have been established to control bacterial wilt [19] because the pathogen *R. solanacearum* can persist for many years in soil, water, weeds, and diseased plant tissues [8]. To avoid chemical contamination in the soil, biological control methods have been rapidly developed in recent years [28,29,30,31]. *Pseudomonas* spp., *Bacillus* spp., *Streptomyces* spp., and other bacterial species are promising for controlling bacterial wilt [19]. Interestingly, several avirulent *R. solanacearum* isolates are able to prevent bacterial wilt disease development [32]. *Pseudomonas* spp. strains are optimal biocontrol agents, as they compete for root colonization, synthesize allele chemicals, and induce systemic resistance of host plants [33]. We recently reported a biocontrol agent, *Pseudomonas* spp. strain A1, with antimicrobial activity against *R. solanacearum* [20]. Despite a remarkable reduction in disease development, the disease index for plants receiving *Pseudomonas* spp. A1 treatment still remained at about 40 at 20 days after inoculation, suggesting that an improved strategy is needed [20].

*Pseudomonas* spp. A1 was initially assigned to the *P*. *putida* group based on a phylogenetic analysis of the 16S rRNA sequence [20]. Based on the complete genome sequence completed in our lab, it was definitively identified as *P*. *mosselii*. Since the initial discovery from clinical specimens in 2002 [34], *P*. *mosselii* isolates have been found extensively in soil and water environments [35,36]. In addition to their ability to degrade a wide variety of aromatic chemicals, *P*. *mosselii* strains show excellent antagonistic activity against plant pathogens [37]. Based on the complete genome sequence information, *P*. *mosselii* A1 studied here possesses a type III secretion system, which is widely distributed in plant-colonizing bacteria, including *P*. *fluorescens* and *P*. *putida* [21]. In *P*. *fluorescens* SBW25, the constitutive expression of *hrpL* and *avrB* confers the ability to induce HR in *A*. *thaliana* ecotype Col-0 [21]. In this study, the hetero-expression of the *R*. *solanacearum ripAA* gene in *P*. *mosselii* A1 resulted in a constitutive expression module and, thus, improved the control efficacy against tobacco bacterial wilt.

*R*. *solanacearum RipAA* acts as an avirulent factor, together with RipP1, to determine the resistance reaction of tobacco to pathogens according to the “gene-for-gene” model [17]. By expressing *ripAA* in *P*. *mosselii* A1, ethylene and jasmonate signaling pathways in tobacco plants were significantly induced relative to the levels in plants treated with *P*. *mosselii* A1. Up to now, the molecular mechanism of hypersensitive response caused by RipAA in tobacco is still unknown. The transcriptome results indicated that these defense pathways are likely to be induced by RipAA. Additionally, further research on the mechanism of RipAA will focus on these differentially expressed defense genes. A previous study reported that the knockout of Haem peroxidase prevents *P*. *putida* to induce systemic resistance [38]. As Haem peroxidase is also found in *P*. *mosselii* A1, we proposed that the wild-type A1 strain possesses the ability to induce systemic resistance. Furthermore, a total of 12 Haem peroxidase genes in tobacco plants were upregulated in response to the expression of *ripAA*, suggesting that oxidative reactions were activated in tobacco plants. The expression of *ripAA* in *P*. *mosselii* A1 induced several defense-related signaling pathways to improve tobacco resistance to bacterial wilt caused by *R. solanacearum*. However, the field use of *P*. *mosselii* AA1 still needs to be carefully evaluated, due to the constitutive induction of defense mechanisms that may be accompanied by a permanent metabolic burden for the plants. Although there was no significant difference in plant height and root length among *P*. *mosselii* AA1, A1 and ddH_2_O treated tobaccos in the early stage (Appendix A), no statistical records were conducted on the growth, flowering, and seed set rates of tobaccos in the late stage.

In *R. solanacearum*, the transcription of *ripAA* is under the control of the key regulator HrpB, which is responsible for the initiation of *hrp* genes, required for the type III secretion system assembly [39]. The expression of *hrpB* is induced in plants or in an *hrp*-inducing, nutrient-poor medium [40]. Even though the transcription of *ripAA* requires induction by environmental signals [11], the *ripAA* promoter drove *ripAA* transcription in *P*. *mosselii* A1 cultured in either Kings B or M63 medium. The transcription of *ripAA* in the engineered AA1 strain was detectable by a sensitive RT-PCR analysis. This suggested that the *hrpB* homolog (CSH50_RS09230) in *P*. *mosselii* A1 was sufficient to activate the transcription of *ripAA*. Unfortunately, we failed to detect the epitope-tagged RipAA-myc in the AA1 strain and, thus, could not determine whether it was delivered to the extracellular milieu. We speculated that the RipAA protein was not present in sufficient quantities for a Western blot analysis. In addition, different concentrations of A1 and AA1 were inoculated into tobacco leaves, respectively. Although the inoculated leaves had no phenotype, the leaves inoculated with high concentrations of *P*. *mosselii* AA1 displayed cell necrosis via trypan blue staining, while the leaves inoculated with the same concentration of *P*. *mosselii* A1 did not (Appendix A). This result also showed that RipAA could be expressed and secreted in *P*. *mosselii* AA1. Irrespective of this, the engineered *P*. *mosselii* AA1 strain expressing *ripAA* improved plant resistance to bacterial wilt.

Our original aim was to create an engineered strain possessing both antimicrobial activity and defense induction abilities to prevent bacterial wilt disease at diverse infection stages. At the invasion stage, *P*. *mosselii* AA1 inhibited *R*. *solanacearum in* the rhizosphere; it also promoted plant defense against the pathogen after colonization in the vascular system. Even though the disease index decreased by about 9.1%, the engineered *P*. *mosselii* AA1 did not fully prevent disease occurrence. There are two explanations for this observation. First, the insertion of *ripAA* in the engineered *P*. *mosselii* AA1 strain did not enhance antimicrobial activity against *R. solanacearum*, which could not lead to a decreased pathogen population in the soil. Second, the incompatible interaction of *R. solanacearum* with tobacco plants is determined by both RipAA and RipP1 [16]. The expression of only the *ripAA* gene in *P*. *mosselii* AA1 could not induce sufficient resistance to bacterial wilt in tobacco plants. Therefore, in a future study, more *Ralstonia solanacearum* effectors acting as avirulence factors in tobacco, such as RipP1 and RipB [16,41], will be expressed or a strong promoter will be used to produce more RipAA in *P*. *mosselii* AA1 to improve the efficacy against tobacco bacteria wilt.

## 5. Conclusions

We constructed an engineered *P*. *mosselii* strain AA1 expressing the *ripAA* gene from *R. solanacearum*, which is involved in incompatible interactions with tobacco plants. *P*. *mosselii* AA1 exhibited improved biological control efficacy against tobacco bacterial wilt by evoking several defense signaling pathways. These results not only provide insight into defense mechanisms induced by *ripAA* but also provide a basis for improving the management of bacterial wilt disease using pathogen-derived Avr proteins.

## Figures and Tables

**Figure 1 biology-11-01170-f001:**
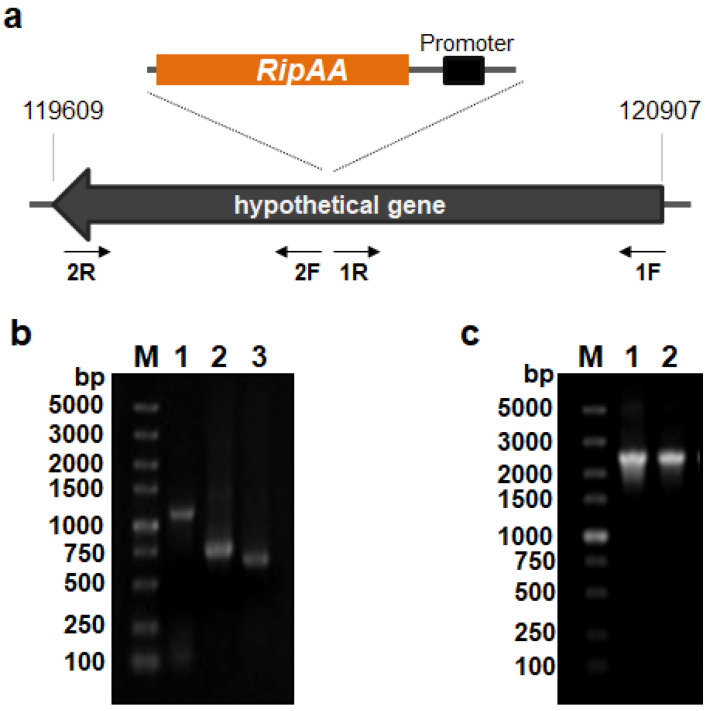
Construction and verification of the engineered *P. mosselii* strain AA1: (**a**) schematic of the strategy to generate a recombinant AA1 strain carrying *ripAA*. The primer sets used for homologous recombination are indicated by arrows. The insertion site of *ripAA* is shown by dotted lines; (**b**) PCR products cloned in the suicide vector pK19mobSacB. Lane M shows a 5 kb DNA marker, lane 1 shows the PCR product *avrA* from *R. solanacearum*, and lanes 2 and 3 show the PCR products obtained using primer sets 1F/1R and 2F/2R from *P. mosselii* A1, respectively; (**c**) PCR verification of engineered *P. mosselii* AA1. Lane M shows a 5 kb DNA marker, and lanes 1 and 2 show the PCR products obtained using the primer set 1F/2R from the recombinant vector and *P. mosselii* AA1 genomic DNA. The product in lane 2 was sequenced to confirm that *ripAA* was inserted at the desired location.

**Figure 2 biology-11-01170-f002:**
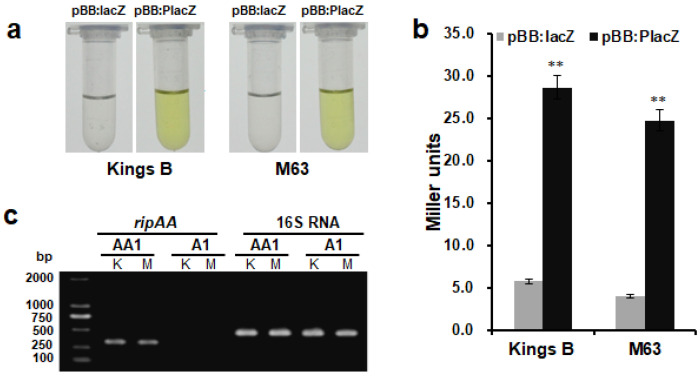
Transcription of *ripAA* in *P. mosselii* AA1: (**a**) yellow color indicates β-galactosidase activity. The *ripAA* promoter was fused to *lacZ* in the pBBR1MCS-5 vector. The pBBR1MCS-5 vector with only *lacZ* was used as a negative control. After both constructs were introduced into *P. mosselii* A1, β-galactosidase activity was assayed in Kings B or M63 broth; (**b**) quantitative analysis of β-galactosidase activity driven by the *ripAA* promoter in *P. mosselii* A1. Error bars represent the standard deviation from three independent experiments. Differences were evaluated using Student’s *t*-tests (** *p* < 0.01); (**c**) semi-quantitative RT-PCR analysis of *ripAA* transcription in *P. mosselii* AA1. Total RNAs were isolated from cells grown in Kings B and M63 broths, and 16S rRNA was used as an internal control. DNA marker was DL2000 (TAKARA, Otsu, Japan). K: Kings B medium. M: M63 medium. Construction and verification of the engineered *P. mosselii* strain AA1.

**Figure 3 biology-11-01170-f003:**
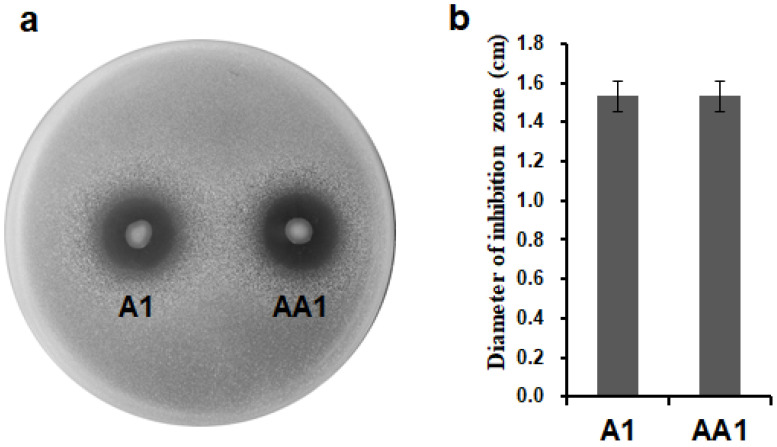
Antimicrobial activity of *P. mosselii* A1 and its mutant against *Ralstonia solanacearum* in a plate confrontation experiment: (**a**) plate confrontation on *Ralstonia solanacearum* by *P. mosselii* AA1. The cultured *P. mosselii* cells were prepared to OD_600_ = 1.5, and 2 µg of cell suspension was spotted on NA plates containing *R. solanacearum.* The inhibitory effect was recorded at 3 days post-inoculation; (**b**) diameters of inhibitory zones, as determined at 3 days post-inoculation from three replications. Error bars represent the standard deviation from three independent experiments.

**Figure 4 biology-11-01170-f004:**
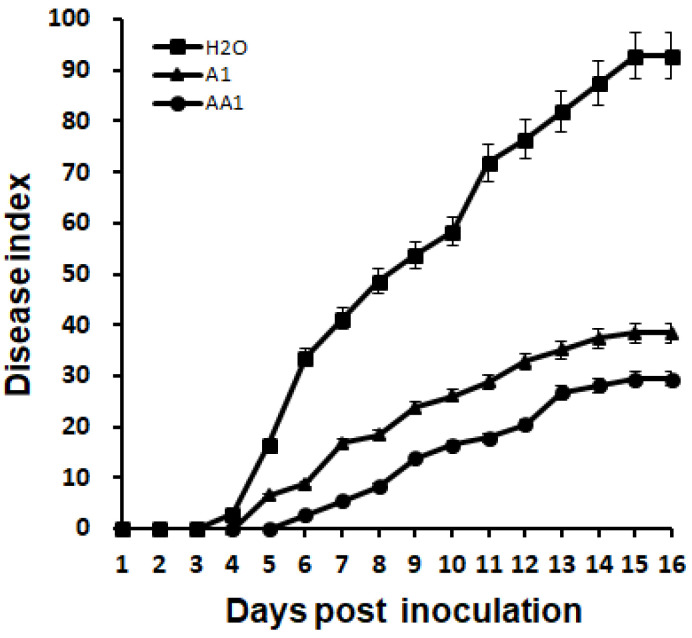
Progression of bacterial wilt on tobacco plants treated with AA1. The disease index was recorded daily from 0 to 16 days after wounded root inoculation. Each point represents the mean disease rate of 8 inoculated plants per treatment. Error bars represent the standard deviation from three independent experiments.

**Figure 5 biology-11-01170-f005:**
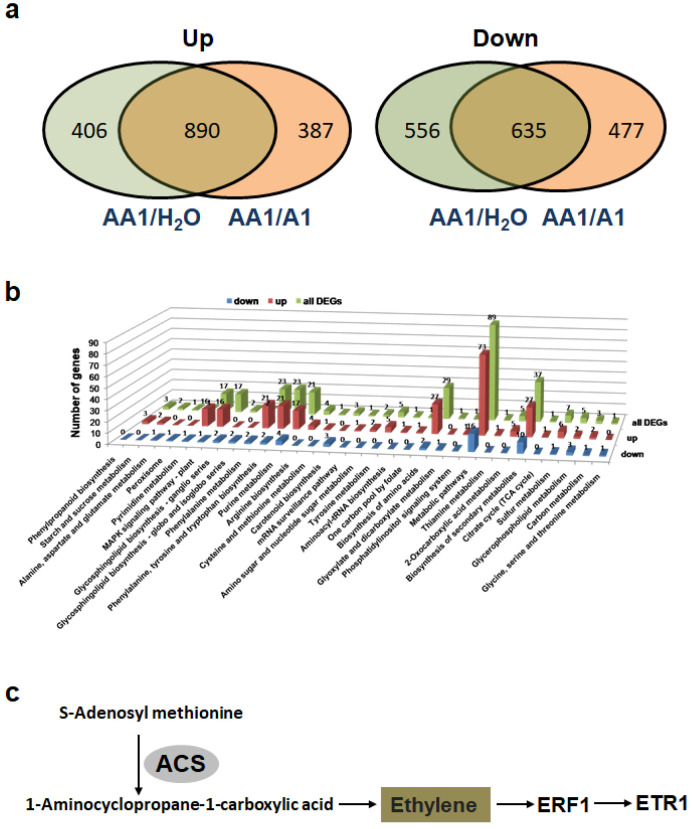
Identification of differentially expressed genes by RNA sequencing: (**a**) diagram showing the numbers of genes that shared the same patterns of transcriptional change. Genes with log2 fold change ≥ 1 and P ≤ 1 for the comparison between control and treated groups were determined using the DESeq2 package in R (version 3.3.2) and Cuffdiff v.2.2.156; (**b**) KEGG classification of differentially expressed genes; (**c**) induced genes involved in ethylene signaling pathway.

**Table 1 biology-11-01170-t001:** Bacterial strains and plasmids used in this study.

Strains or Plasmids	Relevant Characteristics	Resources
**Strains**		
*Pseudomonas mosselii*		
A1	Wild-type *Pseudomonas mosselii* isolated from soil	[20]
AA1	*Pseudomonas mosselii* expressing *ripAA* gene of *Ralstonia solanacearum*	This study
A1/pBB:lacZ	Gm^r^, *Pseudomonas mosselii* A1 harboring pBB:lacZ	This study
A1/pBB:PlacZ	Gm^r^, *Pseudomonas mosselii* A1 harboring pBB:PlacZ	This study
*Escherichia coli*		
DH5α	*F’*Φ*80dlacZDM15D(lacZYA-argF)U169 endA1 deoR recA1 hsdR17(rK2 mK+) phoA supE44 l2 thi-l gyrA96 relA1*	Clontech
*Ralstonia solanacearum*		
RsT1	PB^r^, a *Ralstonia solanacearum* strain isolated from tobacco plants in Sanming, Fujian, China	Lab collection
**Plasmids**		
pK18mobsacB	Km^r^, suicide vector, *sacB*^+^	[24]
pK18:RipAA	Km^r^, a 2.3 kb fusion containing 5′ and 3′ terminal sequences of a hypothetical gene in *Pseudomonas mosselii* A1, which were interspaced by 1.1 kb DNA fragment of *ripAA* gene with native promoter	This study
pBBR1MCS-5	Gm^r^, 4.7 kb broad-host range plasmid, *lacZ*	[25]
pLacZ-Basic	Ap^r^, 7.5 kb pUC replication origin plasmid carrying a β-galactosidase gene	Clontech
pBB:lacZ	Gm^r^, pBBR1MCS-5 harboring a 4.6 kb *Kpn*I/*Sal*I fragment that contains a β-galactosidase gene from pLacZ-Basic	This study
pBB:PlacZ	Gm^r^, pBBR1MCS-5 harboring a 276-bp *ripAA* promoter fused with the 4.6 kb *Kpn*I/*Sal*I fragment from pLacZ-Basic	This study

## Data Availability

All the data of transcriptome were deposited with NCBI Sequence Read Archive (SRA) (https://www.ncbi.nlm.nih.gov/sra (accessed on 19 November 2018)).

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
