# Peer review of "Expression of the ripAA Gene in the Soilborne Pseudomonas mosselii Can Promote the Control Efficacy against Tobacco Bacterial Wilt"

_biology, 2022, doi:10.3390/biology11081170_

Round 1

Reviewer 1 Report

            The paper by Tao et al. describes the genetic modification of Pseudomonas mosselii to express, under the regulation of its native promoter, the ripAA gene from Ralstonia solanacearum, which encode an avirulent factor. The ms. includes assays to evaluate the activity of the ripAA promoter through the expression of the LacZ enzyme, and RT-PCR experiments to demonstrate the transcription of the ripAA gene in the modified P. mosselii AA1 when growing in culture media. The biological activity of P. mosselii AA1 against Ralstonia solanacearum is tested in a plate confrontation experiment, whereas in the plant, P. mosselii AA1 delayed and reduced the wilt disease symptoms caused by R. solanacearum. Finally, RNAseq analysis of plants in the challenge assay with P. mosselii strains A1 and AA1

Altogether, the study is well organized and the experiments are well described, but there are some items not satisfactorily or could be better addressed:

1.      The epitope-tagged RipAA-myc in the AA1 strain was not detected likely because was expressed at a very low amount. I know that the activity of the promoter was tested using an enzymatic assay and the mRNA of ripAA was detected, but I wonder why this protein could not be detected. It is just curiosity!

2.      Although the RNAseq assay provided interesting results, additional experimental supports to validate the analysis were not provided., e.g., RT-qPCR with five to ten of the up- and down-regulated genes in sum.

3.      Is there any experimental support to collect the root samples 2 days after inoculation with the cell suspension of P. mosselii A1 or AA1??

4.      Plant responses to the presence of P. mosselii AA1, as evaluated by the RNAseq study, are consistent with those observed during R. solanacearum- tabaco interaction? In my opinion, this point deserves a better argumentation in the discussion.  

Some other points and recommendations have been detailed below.

Line 94: “The first 604-bp of the DNA fragment containing the N terminus of a hypothetical gene” Replace “N terminus” with 5’-end”

Line 97: “the 677-bp C-terminal region of the hypothetical gene was!”. Replace “C-terminal region” with 3’-end.

Author Response

Response to Reviewer #1:

Reviewer 1:

Altogether, the study is well organized and the experiments are well described, but there are some items not satisfactorily or could be better addressed:

1. The epitope-tagged RipAA-myc in the AA1 strain was not detected likely because was expressed at a very low amount. I know that the activity of the promoter was tested using an enzymatic assay and the mRNA of ripAA was detected, but I wonder why this protein could not be detected. It is just curiosity!

Our Response: The epitope-tagged RipAA-myc in the AA1 strain was not detected, which is a real pity and a problem that puzzle us. In our other studies, RipAA was recombined into the Ralstonia solanacearum genome under its native promoter to form a complementary strain, which also could not be detected by Western blotting. But it could be detected if it was expressed by a complementary plasmid in the genome. And the qPCR results demonstrated that the complementary plasmid inserted into the genome showed a form of overexpression, and the transcription level of ripAA was more than 30 times or even higher than the former. And in this study, RipAA was inserted into the Ralstonia solanacearum genome under its native promoter by homologous recombination and the transcription level of ripAA did not show overexpression. In addition, we inoculated AA1 strain in tobacco leaves, and only high concentration of AA1 strain could induce the cell death in tobacco leaves, which also proved that RipAA could be expressed. Therefore, in our opinion, the reason why the epitope-tagged RipAA-myc in the AA1 strain cannot be detected is that the expression level is at a very low amount. We have added the result of inoculating AA1 strain in tobacco leaves to the revised manuscript. (Line 381-386)

2. Although the RNAseq assay provided interesting results, additional experimental supports to validate the analysis were not provided., e.g., RT-qPCR with five to ten of the up- and down-regulated genes in sum.

Our Response: Thanks for the reviewer’s comments. We have complied with this suggestion and selected ten genes representing different pathways from 1525 differentially expressed genes for qPCR detection. The result showed that the differential expression trend was consistent with the RNAseq assay. And we have added this result in the revised manuscript. (Line 293-295)

3. Is there any experimental support to collect the root samples 2 days after inoculation with the cell suspension of P. mosseliiA1 or AA1??

Our Response: According to our previous works, the GFP-labelled P. mosselii A1 adhered to the wounded root at 2 days post inoculation and was visualized with confocal laser scanning microscopy. The relevant results have been published in the Biological Control (Sun, D.; Zhuo, T.; Hu, X.; Fan, X.; Zou, H. Identification of a Pseudomonas putida as biocontrol agent for tomato bacterial wilt disease. Biol Control. 2017, 114, 45-50.).

4. Plant responses to the presence of P. mosseliiAA1, as evaluated by the RNAseq study, are consistent with those observed during R. solanacearum- tabaco interaction? In my opinion, this point deserves a better argumentation in the discussion.

Our Response: The model pathogen Ralstonia solanacearum GMI1000 is the causal agent of the bacterial wilt disease that attacks many solanaceous plants and other hosts but not tobacco (Nicotiana spp.). GMI1000 possesses around 72 effectors secreted by the type â…¢ secretion system, and several effectors can induce HR in tobacco. It was found that the two effectors RipAA and RipP1 determine the interactions between GMI1000 and tobacco, and RipAA is a major determinant recognized by Nicotiana tabacum. So far, the molecular mechanism of how RipAA is recognized by tobacco have not been reported. Therefore, the RNAseq study will provide an important reference for our next study on the mechanism of RipAA induced HR in tobacco. We have complied with this suggestion and have added this point in the discussion of revised manuscript. (Line 353-356)

Some other points and recommendations have been detailed below.

Line 94: “The first 604-bp of the DNA fragment containing the N terminus of a hypothetical gene” Replace “N terminus” with 5’-end”

Our Response: We have complied with this suggestion and replaced “N terminus” with “5’-end”. (Line 95)

 Line 97: “the 677-bp C-terminal region of the hypothetical gene was!”. Replace “C-terminal region” with 3’-end.

Our Response: We have complied with this suggestion and replaced “C-terminus” with “3’-end”. (Line 98)

Reviewer 2 Report

This paper describes an interesting approach to increase biocontrol of tobacco bacterial wilt by Pseudominas mosselii. By introducing the ripAA gene from Ralstonia solanacearum into P. mosselii, the authors try to induce plant defence mechanisms to enhance biocontrol activity by the microorganism.

The paper is nicely written, with a comprehensive introduction, a clear explanation of the rationale and the experimental approaches, and a good discussion. Apparently, the authors managed to increase disease resistance of tobacco plants more with their transgenic AA1 bacterial strain than with the wildtype A1 strain, and thereby to improve biocontrol.

General comments

When reading the manuscript, I had one problem that prevented me from fully judging the significance of the results. I found it difficult to understand the methodology used to score disease susceptibility / resistance of the plants.

The current manuscript reports the "disease index" of the plants on a scale of 1-100 (Figure 4). As reference for the "disease index" reference [20] is cited (line 146). However, Sun et al (2017) describe the "disease index" as a scale of 0 to 4. But this reference mentions a "Percent severity index" that is calculated using the method described by Ayana et al (2011). So I had to go to a third reference to understand the scoring. According to Ayana et al 2011, "Percent severity index (PSI)" was calculated using the method described by Cooke (2006). So one more reference to go to - and this one is in a book that I do not have access to. So I am unfortunately unable to judge the main result of this paper.

It is important to clarify this methodology for the reader, so that the meaning and significance of the results can be appreciated. I am sure that this can be done very easily with a few adaptions to the manuscript. Maybe the authors were unaware that the methods might not be obvious to readers not expert in this field. I suggest that the authors clarify this directly in the materials and methods section of the present paper with a short description of the index (not only with references to previous publications).

Discussion: I was wondering whether constitutive induction of defence mechanisms in the plant by exposure to the AA1 bacterial strain might increase resistance, but harm plant growth / productivity, so that there is no real advantage for plant performance (or even a negative effect)? You show clearly with your transcriptome analysis that AA1 induces a number of pathogen responses in the plant. This might suggest a permanent metabolic burden for the plants. Did the authors observe phenotypic changes or growth defects for plants treated with the AA1 strain (maybe in control experiments without challenge by Ralstonia solanacearum?). If there are any data or observations available on this, it would be helpful. If not, this possibility should at least be discussed.

Specific comments

·      Line 146: please briefly describe how the disease index is calculated, so the reader can understand the relevance of this central assay without having to go to the literature. Also make clear here and elsewhere in the paper if you talk about the "disease index" (scale 0-4) or the "Percent severity index (PSI)" (scale presumably 0 - 100). 

·      Line 253 ("The plant disease index for the water treatment control increased sharply to 92.9 at 15 days post-inoculation): To me "sharply" sounds as if there is a sudden increase at day 15. When looking at Figure 4, it seems to be a steep, but more or less even increase from day 4 to the end of the experiment. Of course, the increase is steeper in the control than with the treated plants.

·      Lines 254/255: "The disease index for A1 treatment decreased...": "decreased" to me sounds as if the index was higher before (for one set of plants) and then goes down again. Of course you want to express that the disease index was lower for the treated plants than for the controls. I suggest avoiding "decreased" here to prevent misunderstandings. 

·      Line 256/257: “the disease index was reduced 9.1% at 15 days post-inoculation”: It would help the reader and strengthen the argument, if this value could be put more into perspective. What exactly does that mean? You say that first symptoms appear one day later with the AA1 treatment. From the Fig. 4, it looks as if at day 16 the AA1 treated plants have reached a disease index (around 30%) that the A1 plants had reached already five days earlier – so could one speak of a five-day delay in symptom progression (or is it impossible to compare because the plants grow too much during this time, so that one should not compare plants from different days)? Or could the difference be described in another easy to understand way (maybe just for a single timepoint) in addition to using the disease index?

·      Line 259/ Figure 4: In my copy of the manuscript, it is a bit difficult to see the difference between the labels on the graph (circles, triangles, squares), also because of the error bars. Maybe the thickness of the lines could be decreased a bit, or the size of the labels increased?

·      Line 331/332:"A previous study has reported that the knockout of Haem peroxidase enables 331 P. putida to induce systemic resistance [37]". I understand this reference (Matilla et al. 2010) to show that the knockout of the Haem peroxydase prevents induction of the induced systemic resistance?

·      Line 354 ("...P. mosselii AA1 inhibited R. solanacearumin..."): should be "R. solanacearum in...")

All in all, a few minor adaptations to the text would strengthen this interesting paper even further.

Author Response

Response to Reviewer #2:

Reviewer 2:

When reading the manuscript, I had one problem that prevented me from fully judging the significance of the results. I found it difficult to understand the methodology used to score disease susceptibility / resistance of the plants.

The current manuscript reports the "disease index" of the plants on a scale of 1-100 (Figure 4). As reference for the "disease index" reference [20] is cited (line 146). However, Sun et al (2017) describe the "disease index" as a scale of 0 to 4. But this reference mentions a "Percent severity index" that is calculated using the method described by Ayana et al (2011). So I had to go to a third reference to understand the scoring. According to Ayana et al 2011, "Percent severity index (PSI)" was calculated using the method described by Cooke (2006). So one more reference to go to - and this one is in a book that I do not have access to. So I am unfortunately unable to judge the main result of this paper.

It is important to clarify this methodology for the reader, so that the meaning and significance of the results can be appreciated. I am sure that this can be done very easily with a few adaptions to the manuscript. Maybe the authors were unaware that the methods might not be obvious to readers not expert in this field. I suggest that the authors clarify this directly in the materials and methods section of the present paper with a short description of the index (not only with references to previous publications).

Our Response: Thanks for the reviewer’s comments. Bacterial wilt progress was rated daily according to the following the five-point rating scale (0-4) (Sun et al., 2017), where: 0=no wilt symptoms, 1=1 to 25% leaves wilted, 2=26 to 50% leaves wilted, 3=51 to 75% leaves wilted; and 4=76 to 100% leaves wilted. Disease index indicated percent severity index (PSI) and PSI was calculated according to the formula: PSI=∑(each rating scale × number of plants rated in corresponding scale) ×100/(number of all plants rated × maximum scale of the scores) (Cooke, 2006). We have added a description of the index in the revised manuscript. (Line 150-156)

Discussion: I was wondering whether constitutive induction of defence mechanisms in the plant by exposure to the AA1 bacterial strain might increase resistance, but harm plant growth / productivity, so that there is no real advantage for plant performance (or even a negative effect)? You show clearly with your transcriptome analysis that AA1 induces a number of pathogen responses in the plant. This might suggest a permanent metabolic burden for the plants. Did the authors observe phenotypic changes or growth defects for plants treated with the AA1 strain (maybe in control experiments without challenge by Ralstonia solanacearum?). If there are any data or observations available on this, it would be helpful. If not, this possibility should at least be discussed.

Our Response: AA1 bacterial strain expresses the effector protein RipAA, which is considered to have the function of an avirulence gene. Whether it is harmful to plant growth/ productivity is a very important issue. We planted the germinated tobacco seedlings in the soil inoculated with A1, A1 or watered. After 6 weeks, there was no significant difference in plant height and root length of these tobaccos. However, we did not observe whether the aging, flowering and seed set rate of tobacco were affected. In our other studies, there was no significant difference between ripAA transgenic Nicotiana tabacum and wild type. However, this part of the work has not been published, so it is not shown in this article. In future research, we will pay attention to the phenotypic changes or growth defects of plants inoculated AA1 bacterial strain. We have added the discussion in the revised manuscript. (Line 363-369)

Specific comments

  • Line 146: please briefly describe how the disease index is calculated, so the reader can understand the relevance of this central assay without having to go to the literature. Also make clear here and elsewhere in the paper if you talk about the "disease index" (scale 0-4) or the "Percent severity index (PSI)"(scale presumably 0 - 100). 

Our Response: We have complied with this suggestion and added a description of the index in the revised manuscript. (Line 150-156)

  • Line 253 ("The plant disease index for the water treatment control increased sharply to 92.9 at 15 days post-inoculation): To me "sharply" sounds as if there is a sudden increase at day 15. When looking at Figure 4, it seems to be a steep, but more or less even increase from day 4 to the end of the experiment. Of course, the increase is steeper in the control than with the treated plants.

Our Response: We have complied with this suggestion and deleted “sharply” from the sentence in the revised manuscript. (Line 271)

  • Lines 254/255: "The disease index for A1 treatment decreased...": "decreased" to me sounds as if the index was higher before (for one set of plants) and then goes down again. Of course you want to express that the disease index was lower for the treated plants than for the controls. I suggest avoiding "decreased" here to prevent misunderstandings. 

Our Response: We have complied with this suggestion and replaced the first and the second “decreased” with “finally developed” and “developed” in the revised manuscript, respectively. (Line 272-273)

  • Line 256/257: “the disease index was reduced 9.1% at 15 days post-inoculation”: It would help the reader and strengthen the argument, if this value could be put more into perspective. What exactly does that mean? You say that first symptoms appear one day later with the AA1 treatment. From the Fig. 4, it looks as if at day 16 the AA1 treated plants have reached a disease index (around 30%) that the A1 plants had reached already five days earlier – so could one speak of a five-day delay in symptom progression (or is it impossible to compare because the plants grow too much during this time, so that one should not compare plants from different days)? Or could the difference be described in another easy to understand way (maybe just for a single timepoint) in addition to using the disease index?

Our Response: Throughout the experiment, the disease index for AA1 treatment was always lower than A1 treatment. Finally, the disease index for AA1 treatment stayed at 29.4, while A1 treatment stayed at 38.5. So, we said that “the disease index was reduced 9.1% at 15 days post-inoculation”. We have complied with this suggestion and replaced “the disease index was reduced 9.1% at 15 days post-inoculation” with “the final disease index was reduced 9.1%” in the revised manuscript. (Line 275)

  • Line 259/ Figure 4: In my copy of the manuscript, it is a bit difficult to see the difference between the labels on the graph (circles, triangles, squares), also because of the error bars. Maybe the thickness of the lines could be decreased a bit, or the size of the labels increased?

Our Response: We have complied with this suggestion and replaced a new figure with the old one in the revised manuscript. And the thickness of lines in the new Figure 4 was decreased a bit.

  • Line 331/332:"A previous study has reported that the knockout of Haem peroxidase enables 331 P. putida to induce systemic resistance [37]". I understand this reference (Matilla et al. 2010) to show that the knockout of the Haem peroxydase prevents induction of the induced systemic resistance?

Our Response: We have complied with this suggestion and replaced “enables” with “prevents”. (Line 357)

  • Line 354 ("...P. mosselii AA1 inhibited R. solanacearumin..."): should be "R. solanacearum in...")

Our Response: We have complied with this suggestion and replaced “R. solanacearumin…” with “R. solanacearum in…”. (Line 390)

All in all, a few minor adaptations to the text would strengthen this interesting paper even further.

Our Response: Thanks for the reviewer’s comments. Your kind advices are valuable in improving the quality of our manuscript. We have studied comments carefully and have made corrections in the revise manuscript which we hope meet with approval. 

Reviewer 3 Report

The manuscript from Zhuo et al., expressed an avirulence gene ripAA from Ralstonia solanacearum into P. mosselii. Further, they have shown the transcription of this gene into bacterium and tested the antimicrobial activity against Ralstonia solanacearum. Based on transcriptome analysis, authors hypothesized the upregulation and downregulation of genes in defense pathways. The study is interesting and has important implications towards controlling tobacco bacterial wilt. However, there are many caveats and comments the authors must address as given below:

-       The author focused on one strategy that is expressing a single avirulent gene that seemed to work a bit in terms of disease index but they did not try to improve it further.

-       The authors mentioned potential reasons in discussion why expression of RipAA could not lead to decrease pathogen population such as co-expression of RipAA and RipP1. But they did not try it in this study. From literature it is evident that both RipAA and RipP1 in combination determine the host range of R. solanacearum. In my opinion the authors should try to test/improve disease index with multiple combination instead of relying on a single gene strategy.

-       Another point that authors mentioned was if the RipAA expression was strong enough since the authors could not perform western blot. I would recommend the authors to investigate this bottleneck by enzymatic assay. Although the authors confirmed the expression at transcriptional level, but nothing was done to confirm at translational level if the protein was produced or if it is biologically active.

-    Section 3.2: The LacZ experiment did not establishes the transcription of the ripAA gene but it just test the functionality of the native promoter in Pseudomonas species. The experiment is not required (not contributing much) since the authors already shown the functionality of promoter and transcription of gene doing RT-PCR quantification.

-       Did author, codon optimized the ripAA gene. May be that is the reason for the low protein titers. Alternatively, the authors could try to use alternative promoters to increase the expression.

-       Material and method section needs to be more detailed, like include the media composition, It is not clear if the plasmid was generated in three different cloning steps on in one single step?

-       The section 3.1 and section 2.2 are providing more and less the same information.

-       Had the neutrality of insertion site has been tested, if yes provide reference.

-       The current manuscript is providing the same information which has already been demonstrated in literature. Therefore, limiting the novelty of the study.

Author Response

Response to Reviewer #3:

Reviewer 3:

The manuscript from Zhuo et al., expressed an avirulence gene ripAA from Ralstonia solanacearum into P. mosselii. Further, they have shown the transcription of this gene into bacterium and tested the antimicrobial activity against Ralstonia solanacearum. Based on transcriptome analysis, authors hypothesized the upregulation and downregulation of genes in defense pathways. The study is interesting and has important implications towards controlling tobacco bacterial wilt. However, there are many caveats and comments the authors must address as given below:

- The author focused on one strategy that is expressing a single avirulent gene that seemed to work a bit in terms of disease index but they did not try to improve it further.

Our Response: Thanks for the reviewer’s comments. We initially wanted to express RipAA, the Ralstonia solanacearum model strain GMI1000 effector recognized by tobacco, in biocontrol agent Pseudomonas mosselii A1 to test whether it could improve the efficiency of controlling tobacco bacterial wilt. In future work, we will try to express Ralstonia solanacearum multiple effector proteins recognized by tobacco in AA1 at the same time, such as RipP1, RipB, etc., or replace the promoter of ripAA in AA1 to increase the expression of ripAA, to improve the control efficiency. We have added the discussion of more strategies in the revised manuscript. (Line 398-402)

- The authors mentioned potential reasons in discussion why expression of RipAA could not lead to decrease pathogen population such as co-expression of RipAA and RipP1. But they did not try it in this study. From literature it is evident that both RipAA and RipP1 in combination determine the host range of R. solanacearum. In my opinion the authors should try to test/improve disease index with multiple combination instead of relying on a single gene strategy.

Our Response: According to previous report, both RipAA and RipP1 in combination determined the host range of R. solanacearum on at least three tobacco species, but RipAA was the major determinant recognized by N. tabacum and N. benthamiana, while RipP1 appeared to be the major HR elicitor on N. glutinosa. And N. tabacum is mostly planted in production, so only RipAA were selected to be express in Pseudomonas mosselii A1. But in further work, we will co-express RipAA and RipP1 in Pseudomonas mosselii A1. We have added the discussion of this strategy in the revised manuscript. (Line 398-402)

- Another point that authors mentioned was if the RipAA expression was strong enough since the authors could not perform western blot. I would recommend the authors to investigate this bottleneck by enzymatic assay. Although the authors confirmed the expression at transcriptional level, but nothing was done to confirm at translational level if the protein was produced or if it is biologically active.

Our Response: Thanks for the reviewer’s comments. It’s a great pity that the RipAA expression could not detected by Western blotting. We have refered to the relevant literature and found that calmodulin-dependent adenylate cyclase domain (Cya) of the Bordetella pertussis cyclolysin could be used as a reporter gene to be fused with RipAA, and then, the expression of RipAA can be detected by cAMP concentration detection assay. Due to the imperfect experimental system and time constraints, the experiment failed. But we will continue to work hard and try to establish this method in future research. Meanwhile, we inoculated A1 and AA1 strains in tobacco leaves and found that only high concentration of AA1 strain could induce the necrosis of tobacco cells, which might provide a proof that RipAA could be produced and was biologically active. We have added the result of inoculating AA1 strain in tobacco leaves to the revised manuscript. (Line 381-386)

- Section 3.2: The LacZ experiment did not establishes the transcription of the ripAA gene but it just test the functionality of the native promoter in Pseudomonas species. The experiment is not required (not contributing much) since the authors already shown the functionality of promoter and transcription of gene doing RT-PCR quantification.

Our Response: At first, we were not sure whether the native promoter of RipAA could work in Pseudomonas mosselii A1, so we first used the LacZ experiment to prove that this promoter could work normally in A1. Then, the promoter linked ripAA gene was homologously recombined into the genome of A1, and finally the transcription of the ripAA gene were detected. Though the experiment might not contribute much, the result enhanced the reliability of the transcription expression of ripAA.

- Did author, codon optimized the ripAA gene. May be that is the reason for the low protein titers. Alternatively, the authors could try to use alternative promoters to increase the expression.

Our Response: Thanks for the reviewer’s comments. This is a good strategy to increase the expression of ripAA and we will try to optimize condons of ripAA and use a stronger promoter in further work. We have added the discussion of this strategy in the revised manuscript. (Line 398-402)

- Material and method section needs to be more detailed, like include the media composition, It is not clear if the plasmid was generated in three different cloning steps on in one single step?

Our Response: We have complied with this suggestion and supplemented the media composition (Line 84-85, 87) and described the generation of plasmid in detail in the revise manuscript. (Line 117-124)

- The section 3.1 and section 2.2 are providing more and less the same information.

Our Response: We agree that the section 3.1 and section 2.2 are providing more and less the same information, but section 2.2 mainly describes the detail of construction process of the plasmids and recombinant strains, and section 3.1 mainly shows the construction strategy and results.

- Had the neutrality of insertion site has been tested, if yes provide reference.

Our Response: The neutrality of insertion site has not been tested. The genes inserted by ripAA were selected according to the annotation of Pseudomonas mosselii A1 genome. The reason for selection was that he insertion of this site would not affect the growth, reproduction and biocontrol characteristics of A1.

-  The current manuscript is providing the same information which has already been demonstrated in literature. Therefore, limiting the novelty of the study.

Our Response: The expression of effector that determine the incompatible interaction between Ralstonia solanacearum and tobacco in biocontrol agents to improve the efficacy against tobacco bacterial wilt has not been reported. So far, the molecular mechanism of how RipAA is recognized by tobacco have not been reported. Therefore, our RNAseq study will provide an important reference for our next study on the mechanism of RipAA induced HR in tobacco. (Line 353-356, 407-409)

Round 2

Reviewer 2 Report

The authors have addressed all my concerns adequately and have introduced some additional clarifications.

Reviewer 3 Report

The authors have provided adequate response to the concerns raised by me in my previous review. I am satisfied by the author's response and therefore endorse the manuscript for publication in its current form.